# Comparison between Immunocytochemistry, FISH and NGS for *ALK* and *ROS1* Rearrangement Detection in Cytological Samples

**DOI:** 10.3390/ijms231810556

**Published:** 2022-09-12

**Authors:** Diane Frankel, Isabelle Nanni, L’Houcine Ouafik, Clara Camilla, Eric Pellegrino, Nathalie Beaufils, Laurent Greillier, Hervé Dutau, Philippe Astoul, Elise Kaspi, Patrice Roll

**Affiliations:** 1Aix Marseille Univ, APHM, INSERM, MMG, Hôpital la Timone, Service de Biologie Cellulaire, 13005 Marseille, France; 2APHM, Faculté de Médecine Nord, Service d’Oncobiologie, 13015 Marseille, France; 3Aix Marseille Univ, CNRS, INP, Inst Neurophysiopathol, 13005 Marseille, France; 4Multidisciplinary Oncology and Therapeutic Innovations, Marseille University Hospital (APHM), Aix Marseille University, 13015 Marseille, France; 5Department of Thoracic Oncology, Pleural Diseases and Interventional Pulmonology, APHM, 13015 Marseille, France

**Keywords:** immunocytochemistry, FISH, NGS, ALK, ROS1, lung cancer, adenocarcinoma, cytology

## Abstract

The detection of *ROS1* and *ALK* rearrangements is performed for advanced-stage non-small cell lung cancer. Several techniques can be used on cytological samples, such as immunocytochemistry (ICC), fluorescence in situ hybridization (FISH) and, more recently, next-generation sequencing (NGS), which is gradually becoming the gold standard. We performed a retrospective study to compare *ALK* and *ROS1* rearrangement results from immunocytochemistry, FISH and NGS methods from 131 cytological samples. Compared to NGS, the sensitivity and specificity of ICC were 0.79 and 0.91, respectively, for ALK, and 1 and 0.87 for ROS1. Regarding FISH, the sensitivity and specificity were both at 1 for *ALK* and *ROS1* probes. False-positive cases obtained by ICC were systematically corrected by FISH. When using ICC and FISH techniques, results are very close to NGS. The false-positive cases obtained by ICC are corrected by FISH, and the true-positive cases are confirmed. NGS has the potential to improve the detection of *ALK* and *ROS1* rearrangements in cytological samples; however, the cost of this technique is still much higher than the sequential use of ICC and FISH.

## 1. Introduction

Lung cancer is the most frequently diagnosed cancer across the world, and the leading cause of cancer death in both men and women [1]. Non-small cell lung cancer (NSCLC) represents 85% of lung cancers, among which 40% are adenocarcinomas. The emergence of tyrosine kinase inhibitors (TKI) has changed the diagnosis and therapeutic management of NSCLC. When an advanced stage is diagnosed, *EGFR* mutational status and *ALK* and *ROS1* rearrangements are now routinely performed [2]. *ROS1* and *ALK* rearrangements account, respectively, for 2–3% and 3–7% of non-small cell lung cancer cases [3,4,5,6,7]. The classical method for screening these rearrangements is immunohistochemistry (IHC). Positive results need to be confirmed by fluorescent in situ hybridization (FISH) for *ROS1*, regardless of the intensity level, and for *ALK* when the positivity is less than 3+ [8,9,10]. Next-generation sequencing (NGS) examines several cancer–driver–gene alterations, thereby providing a mutational portrait. The European Society for Medical Oncology (ESMO) has published recommendations for the use of NGS in tumors from patients with advanced NSCLC based on the ESMO Scale for Clinical Actionability of molecular Targets (ESCAT) [11]. NGS can be performed on DNA for mutational status (i.e., for *EGFR*, *BRAF*, *MET*, *KRAS*, *ERBB2*, *BRCA1/2* and *PIK3CA*) and on RNA for the detection of fusion transcripts (i.e., for *ALK*, *ROS1*, *NTRK*, *RET* and *NRG1* rearrangements).

As we have previously shown, cytological sample are a pertinent alternative for detecting *ALK* and *ROS1* rearrangements [12,13]. According to the ESMO recommendations, patients diagnosed with advanced NSCLC in Assistance Publique des Hôpitaux de Marseille (APHM) are tested to detect molecular alterations in tumors. In cytological samples, *ALK* and *ROS1* rearrangements were performed by FISH since 2015, immunocytochemistry (ICC) since 2017, and by NGS fusion panel from 2018 onward. In this study, we compared the results of *ALK* and *ROS1* rearrangements on cytological samples obtained by ICC and/or FISH and NGS panel fusion.

## 2. Results

Between November 2018 and December 2021, a total of 142 samples were sent from the Cell Biology Laboratory to the Oncobiology Laboratory for panel fusion testing. Eleven were excluded from this analysis because the diagnosis was neither lung adenocarcinoma nor NSCLC not otherwise specified (NOS). Among the 131 patients included, 73 were males and 58 were females. The average age was 67.4 years old. The majority of the patients were diagnosed with stage IV lung adenocarcinoma (76.3% out of a total of 80.2% of patients with lung adenocarcinoma), and were former or active smokers (70.2%) (Table 1).

Among the 131 samples, 59 (45.0%) were lymph nodes collected by endobronchial ultrasound transbronchial aspiration (EBUS-TBNA), 45 (34.3%) were pleural effusions, nine (6.9%) were mediastinal or pulmonary masses collected by EBUS-TBNA, seven (5.3%) were pericardial effusions, five (3.8%) were bronchial brushings, four (3.1%) were bronchoalveolar lavage fluids (BAL) and two (1.5%) were cerebrospinal fluids (CSF) (Appendix A). For NGS analysis, 114 samples were sent as frozen cell pellets and 17 as stained slides. The RNA extraction failed for eight samples (frozen cell pellets), thus preventing the NGS panel testing. All the 17 slides used allowed extraction of enough RNA to perform the NGS panel. The absence of rearrangement of both *ALK* and *ROS1* was found in 105 samples using the NGS panel fusion, including those from the 17 slides.

### 2.1. NGS Results

*ALK* exon 20 was rearranged in 15 samples and the partner concerned six different exons of *EML4*: *EML4(6)-ALK(20)* in six cases, *EML4(13)-ALK(20)* in four cases, *EML4(2)-ALK(20)* in two cases, *EML4(14)-ALK(20)* in one case, *EML4(20)-ALK(20)* in one case and *EML4(18)-ALK(20)* in one case. These samples concerned nine females and six males, with a mean age of 64.7 ± 14.4 years old. For one patient, two samples were tested 2 months apart. The first one was obtained from a stained slide, and no fusion was detected, whereas the second one obtained from a frozen cell pellet led to identification of *EML4(6)-ALK(20)* fusion.

*ROS1* rearrangements with different fusion partners were found in three samples: *SDC4(4)-ROS1(34)*, *SLC34A2(13)-ROS1(32)* and *CD74(6)-ROS1(34)*. The fusion panel testing revealed two other types of rearrangements: a *MET* rearrangement in one patient and a *RET* rearrangement in another.

### 2.2. Immunohistochemistry and FISH Results

Concerning the ICC technique, 30 samples could not be tested because they had been entirely used for diagnosis and NGS, or because the percentage of malignant cells were <5%, preventing a reliable result. A positive or doubtful result was obtained for 18 samples with ALK antibody and for 15 samples with ROS1 antibody. These samples were tested by FISH, and only 13 were positive for *ALK* rearrangement and two for *ROS1* rearrangement, respectively (Figure 1A–L). It is to note that one of the three samples with *ROS1* rearrangement found by NGS, the result was non-interpretable. In one pericardial effusion, ROS1, was positive in ICC, but the FISH result showed a loss of signal in 66% of the nuclei. In seven samples, ALK and ROS1 were both doubtful with the ICC technique. The FISH showed a gene copy number gain for both probes with at least three signals per probe, probably due to a hyperdiploid karyotype or gene amplification in malignant cells (Figure 1M–R).

### 2.3. ICC and FISH Sensitivity and Specificity

Using NGS as a reference and counting only samples with contributing results for both ICC or FISH techniques, the sensitivity and specificity of ICC were 0.79 and 0.91, respectively for ALK, and 1 and 0.87 for ROS1. Regarding FISH technique, the sensitivity and specificity were both at 1 for *ALK* and *ROS1* probes (Table 2). The concordance between the three methods is shown in the Venn diagram in Figure 2.

### 2.4. Additive Results

In addition to the panel fusion, a second NGS DNA mutation panel was systematically performed. Among 123 samples tested, at least one mutation was found in 94 samples. The most frequent was *TP53* in 60 (48.8%) samples, followed by *KRAS* in 40 (32.5%) samples, *EGFR* in 19 (15.4%), *BRAF* in eight (6.5%), *STK11* in seven (5.7%), *MET* in three (2.4%) samples, *ERBB2* in two (1.6%) samples, as well as *CTNNB1*, *FGFR3*, *PIK3CA*, *RET* and *SMAD4*. Furthermore, *ALK*, *ERBB4*, *PDGFRA* and *POLE* were individually found in one sample (0.8% each). *ERBB2* amplification was found in one (0.8%) sample (Appendix A).

Three patients with *ALK* rearrangement presented co-occurring mutations. One patient had a mutation in *BRAF* and *SMAD4*, one had a mutation *CTNNB1* and one had a mutation in *TP53*. The presence of both *ALK* rearrangement and *TP53* mutation has already been described by Kron et al. [14]. Concerning the three patients with *ROS1* rearrangement, two of them had a mutation in *TP53*. Concerning other mutations, 42 patients had a mutation in *TP53* associated with another mutation, and among them, 29 patients exhibited a *KRAS* mutation (Appendix A).

## 3. Discussion

As cytological samples are often the only samples available for advanced NSCLC diagnosis and biomarker testing, cytology research must establish protocols for *ALK*, *ROS1* and *EGFR* testing [15,16]. The use of cytological samples for the identification of *ALK* and *ROS1* rearrangements, using ICC, FISH and NGS, has been validated and is recommended by expert consensus [2,13,17,18].

In the Cell Biology Laboratory of APHM, we chose to analyze the ICC and FISH results on cytospins rather than on cell blocks. Indeed, the use of cytospins allows one to preserve cellular integrity and circumvents nuclear truncation. We performed the screening of *ALK* and *ROS1* rearrangements by ICC. This procedure is cost-effective and easily integrated into routine diagnosis, and has a quick turnaround of results as the rearrangements are relatively rare. When the result is positive or doubtful, the FISH technique is performed to confirm, or disprove, the rearrangement.

For *ALK* rearrangements, the NGS found 15 positive cases among 123 tested, which represent 12.2%. This result is above those described in the literature using FISH or immunohistochemistry (IHC) [6,7]. Using ICC, we found 18 positive or doubtful results that were confirmed in 13 cases by FISH. However, in three cases, ICC technique was negative whereas *EML4*(13)–*ALK*(20) rearrangement was found by NGS. This fusion type is one of the most commonly identified and was found in 27.3 to 41% of patients presenting *ALK* fusion analyzed by NGS [19,20,21]. A perfect correlation was obtained between NGS and FISH for the 27 cases tested by the two techniques (13 positives and 14 negatives).

Concerning *ROS1*, NGS found three patients presenting a rearrangement representing 2.4%. This is in accordance with the percentage described in the literature using IHC and FISH [3,4,5]. The ICC technique detected 15 positive or doubtful cases, among which only two were confirmed by FISH. For one patient, the result was non-interpretable. In all the false-positive results obtained by ICC, an abnormal copy number was found by FISH (copy number gain on 11 cases and loss of signal in one case). The sensitivity and the specificity of ROS1 IHC compared to FISH have been calculated in several studies with a mean of 0.96 and 0.94, respectively, [2], which is in concordance with the results obtained in our study.

Most studies have compared the results obtained on histological samples between IHC and FISH [12]. The sensitivity and specificity mean of 15 studies representing 3919 samples, are at 0.97 and 0.99, respectively. Clavé et al. compared the results of *ALK* and *ROS1* rearrangements obtained on 38 paraffin-embedded samples from non-small cell lung cancer between FISH and NGS. They found discordance in five samples. Among them, four had an isolated 3′ signal FISH pattern. This discrepancy has also been described by Liu et al., who compared results derived by IHC, FISH and NGS [22]. They found one case of an IHC false-negative result confirmed positive by FISH, but not by NGS. A recent study compared IHC, FISH and NGS to detect *ALK* rearrangement from formalin-fixed paraffin-embedded (FFPE) samples. They found a concordance of 75.9% between IHC and FISH and 87.5% with NGS [23]. In our study, we had strong concordance between results obtained by FISH and NGS.

The false-positive results we obtained by ICC concerned, in most cases, the positivity of both ALK and ROS1. The FISH found an increase of signals for *ALK* and *ROS1* with at least three signals, probably corresponding to a hyperdiploidy or a copy gain number as already described for *ALK* [24].

NGS is a technique with a turn-around time and a cost higher than ICC and FISH but has the advantage of evaluating a panel of gene rearrangements in a single analysis. NGS can identify the partner gene concerned when a rearrangement is found, even for *ALK* rearrangement, the tyrosine kinase inhibitor (TKI) can be prescribed irrespective of the gene fusion partner [2]. The feasibility of using cytological samples to detect gene fusion is relatively recent. Before detecting gene fusion, Baum et al. analyzed *EGFR* and *KRAS* mutations in 30 cytological samples from lung adenocarcinoma patients, either on cell blocks or on smears. The NGS was successful in 90% when more than 100 malignant cells were present in the sample [25]. In our study, NGS was successful in 93.8% of cases. Among 108 malignant pleural effusions, Ruan et al. identified six with *ALK/EML4* rearrangements and one with *ROS1/CD74* rearrangement [26]. Yamamoto et al. tested 111 cytological samples (e.g., pleural effusion, cerebrospinal fluid, ascitic fluid, sputum and pericardial effusion) and obtained a result in 90% of cases. *ALK* was rearranged (*EML4–ALK*) in three of them [27]. The tumor cell quantity is a parameter that has to be considered for the management of molecular testing. We have already shown that a very small number of malignant cells can be sufficient to detect mutations in cerebrospinal fluid from meningeal carcinoma [28].

The cost of the analysis is also a parameter that must be considered. In our institution, the cost of NGS compared to ICC alone is four times higher, and when it is combined with FISH it is still two to three times higher. Schluckebier et al. evaluated the cost effectiveness of sequential testing, first for *EGFR* mutations by RT-PCR and then *ALK* and *ROS1* by FISH, compared to NGS [29]. They showed that the NGS was not cost-effective, and displayed a higher probability of correct diagnoses. However, a Canadian study showed that ROS1 screening by IHC with confirmation by FISH remains less expensive than NGS [30]. NGS is a technology that requires a level of expertise that most of pathologists do not possess. A recent Spanish publication showed that IHC and FISH remain the most widely used techniques compared to NGS, with only 1 out of 44 centers using the latter [31].

## 4. Materials and Methods

### 4.1. Sample Collection

The samples included in this study were obtained from patients attending the Assistance Publique des Hôpitaux de Marseille for diagnosis and treatment. The only inclusion criteria was the availability of a cytological sample for biomarker studies. This project was approved by the local ethics committee (PADS22-31). The conventional cytological diagnosis was performed by the Cell Biology Laboratory, and the molecular testing was performed by the Cell Biology Laboratory (ROS1 and ALK by ICC and FISH) and the Oncobiology Laboratory (NGS). NGS was performed from frozen cell pellets or, alternatively, stained slides if the entire sample was used for the conventional cytological diagnosis. Results of the molecular testing and clinical data were retrospectively analyzed.

### 4.2. Immunocytochemistry

Samples were prepared as previously described [32], following manufacturer’s instructions. At least one wash was performed between each step. Slides were fixed with paraformaldehyde (PAF) 4% for 10 min, and then incubated with the peroxidase-blocking solution for 30 min. After being washed, slides were incubated with SensiTEK HRP Kit (ScyTek) for 10 min. Primary antibodies (ALK antibody: clone 5A4, 1/25, ab17127, Abcam; ROS1 antibody: clone D4D6, 1/250, #3287, Cell signaling) were incubated for 30 min. Then, the biotinylated secondary antibody was incubated for 15 min, followed by Streptavidin/HRP for 20 min and DAB Quanto chromogen for 5 min. Nuclei were counterstained with Mayer’s hemalun solution. Slides were mounted with Aquatex^®^. Negative controls (total non-immune mouse or rabbit IgG as primary antibody) and positive H2228 (*ALK* rearranged, Figure 1S) and HCC78 (*ROS1* rearranged, Figure 1T) cell lines were added to each experiment to validate the results. Slides were observed under optical microscope (Leica, Wetzlar, Germany).

### 4.3. FISH

Slides were incubated for 10 min in HCl 0.1N/pepsin (0.5%) at 37 °C, and then washed in PBS. Slides were incubated in formaldehyde (1%) for 10 min, dehydrated in ethanol 70°, 90° and 100° (3 min each) and dried. Denaturation and hybridization were performed with a thermobrite (Leica Biosystems, Wetzlar, Germany). Denaturation was performed following manufacturer’s instructions: 10 min at 75 °C for *ROS1* probe (zytolight SPEC ROS1 Dual-Color Break-Apart probe (PL101), zytovision, Bremerhaven, Germany), 4 min at 75 °C for *ALK* probe (Vysis LSI ALK Dual-Color Break-Appart, rearrangement probe, Abbott molecular, Des Plaines, IL, USA). Hybridization was performed overnight at 37 °C. Slides were washed in a buffered bath (saline–sodium citrate buffer 0.4Xat 72 °C for 2 min, then twice at room temperature for 30 s). Nuclei were stained with DAPI for 10 min at room temperature (DAPI II, Abbott, Chicago, IL, USA). Slides were mounted using FluorSave™ reagent (Merck Millipore, Burlington, USA) and observed using a fluorescence microscope (Leica, Wetzlar, Germany).

### 4.4. Next-Generation Sequencing

The total nucleic acids were extracted with the Maxwell RSC DNA FFPE Kit (Promega, Madison, USA) for the stained slides or with the Maxwell RSC Cell DNA Kit (Promega) for the frozen cell pellets. The pellet was resuspended in PBS solution at the rate of 1 × 10^6^ cells per 100 μL of PBS before extraction. The nucleic acids were recovered in 70 μL of elution buffer and quantified by real-time quantitative PCR for NGS analysis (screening for mutations and fusion transcripts).

The RNAs were extracted specifically from the Maxwell using the Maxwell RSC Simply RNA Blood Kit (Promega) after re-suspension in 200 µL of homogenization solution/thioglycerol and treatment with DNAse. The RNAs were eluted at 50 μL and quantified by spectrophotometry on the Nanodrop 8000 (Thermo Fisher Scientific, Waltham, MA, USA).

The RNAs (ideally 10 ng) were then reverse transcribed into complementary DNA using the SuperScript IV VILO enzyme (Thermo Fisher Scientific). The detection for rearrangements were carried out using next-generation sequencing (NGS) technology on the Ion Torrent S5XL with the Oncomine Focus RNA assay Kit (Thermo Fisher Scientific) (see Appendix A for the 5× Solid Tumor Fusion Transcript Panel). The analysis was performed with the Ion Reporter software (Thermo Fisher Scientific). A minimum of 50,000 mapped reads was required to allow interpretation of the result.

For mutation screening, we used our custom panel Oncomine Solid Tumor and Oncomine Solid Tumor+ (OST/OST+), which includes hotspots of target regions of 28 genes of interest in oncogenesis (Appendix A). For all clinical samples, we performed sequencing with Ion Torrent S5XL (Thermo Fisher, Bourgoin, France) with a sensitivity of 5% for the minimum coverage of 500X. Then, sequencing data were analyzed through two pipelines. The first pipeline was developed by Thermo Fisher on the Ion Torrent Suite + Ion Reporter. Ion Torrent Suite generates FASTQ data and ensures BAM (binary alignment mapping) alignment with the hg19 reference genome by using the TMAP (Torrent Mapping Alignment Program). Ion Reporter makes variant caller and variant annotations. The second pipeline was developed in our laboratory and runs open-source software such as BWA-MEM for alignment, SAMtools for mpileup, VarScan2 as variant caller and VEP Ensemble for annotations. All data are stored in our local MySQL database.

## 5. Conclusions

Cytological samples are adequate for molecular testing. When using ICC combined with FISH technique, the results are very close to those of NGS for *ALK* and *ROS1* rearrangement detection. The false-positive cases obtained by ICC are corrected by FISH and the true-positive cases are confirmed, which demonstrates high specificity and sensitivity. NGS has the potential to improve the detection of *ALK* and *ROS1* rearrangements in cytological samples, and may become the gold standard for molecular genotyping of patients with advanced non-small cell lung cancer. However, due to the cost and the required expertise of biologists and/or bioinformaticians to use this technique, ICC and FISH are still the most commonly used techniques at the moment.

## Figures and Tables

**Figure 1 ijms-23-10556-f001:**
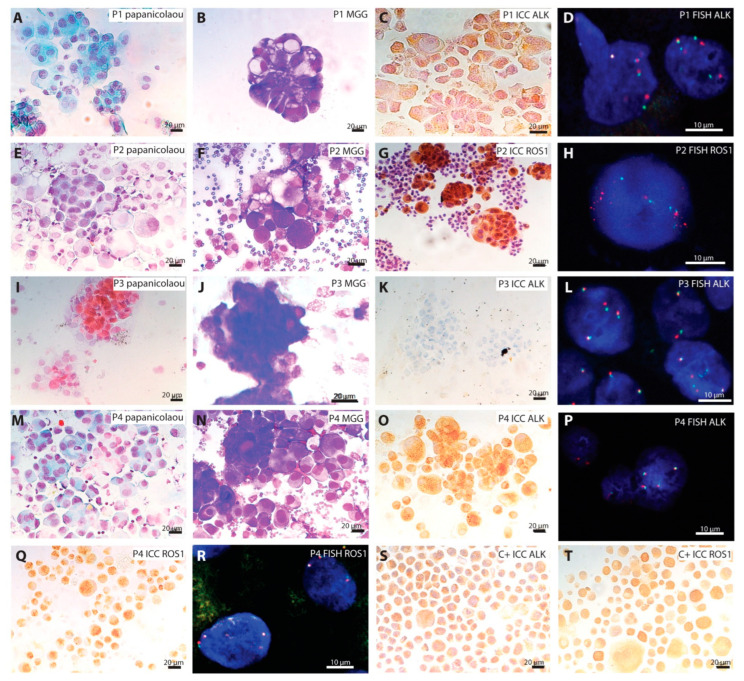
Example of malignant cells with or without *ALK* and *ROS1* rearrangements. (**A**–**D**) Represent the cytology of pleural effusion (P1) containing *ALK* rearranged malignant cells. (**A**) Papanicolaou stain, (**B**) May–Grünwald–Giemsa (MGG) stain, (**C**) immunocytochemistry (ICC) (peroxidase staining) using antibody against ALK (clone 5A4), (**D**) *ALK* FISH probe. (**E**–**H**) Represent the cytology of pleural effusion (P2) containing *ROS1* rearranged malignant cells. (**E**) Papanicolaou stain, (**F**) May–Grünwald–Giemsa stain, (**G**) immunocytochemistry (peroxidase staining) using antibody against ROS1 (clone D4D6), (**D**) *ROS1* FISH probe. (**I**–**L**) Represent the cytology of a lymph node (P3) containing *ALK* rearranged malignant cells. (**I**) Papanicolaou stain, (**J**) May–Grünwald–Giemsa stain, (**K**) immunocytochemistry (peroxidase staining) using antibody against ALK (clone 5A4), (**L**) *ALK* FISH probe. Of note, the false-negative result was obtained with ICC, whereas the rearrangement is observed with FISH. (**M**–**R**) Represent the cytology of pleural effusion (P4) containing malignant cells without *ALK* and *ROS1* rearrangements. (**M**) Papanicolaou stain, (**N**) May–Grünwald–Giemsa stain, (**O**) immunocytochemistry (peroxidase staining) using antibody against ALK (clone 5A4), (**P**) *ALK* FISH probe. (**Q**) Immunocytochemistry using antibody against ROS1 (clone D4D6), (**R**) *ROS1* FISH probe. Of note, the false-positive results were obtained with ICC using antibodies against ALK and ROS1, whereas the FISH shows more than two signals without rearrangement using *ALK* and *ROS1* probes. (**S**) H2228 cell line showing positive staining against anti-ALK antibody (positive control, C+). (**T**) HCC78 cell line showing a positive staining against anti-ROS1 antibody (positive control, C+). Black scale bar represents 20 µm. White scale bar represents 10 µm.

**Figure 2 ijms-23-10556-f002:**
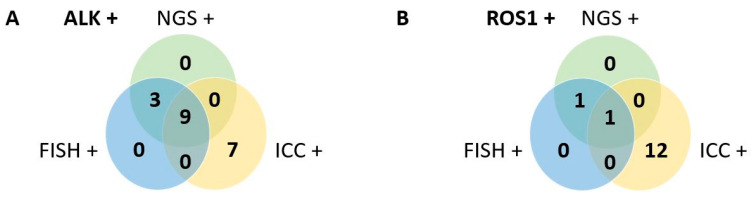
A Venn diagram of ALK-positive (**A**) and ROS1-positive (**B**) cases detected by NGS, ICC and FISH in cytological samples. Four patients and one patient were excluded for ALK or ROS1 results, respectively, because of non-interpretable results in one of the methods.

**Table 1 ijms-23-10556-t001:** Patient demographics.

	Parameter	n (%)
**Age (years)**	Mean ± SD	**67.4 ± 12**
	Range	**36–90**
**Gender**	Male	**73** (55.7)
	Female	**58** (44.3)
**Histopathological type**	Lung adenocarcinoma	**105** (80.2)
	NSCLC NOS	**26** (19.8)
**Smoking status**	Never	**33** (25.2)
	Current/former	**92** (70.2)
	Unknown	**6** (4.6)
**Stage**	I	**2** (1.5)
	II	**5** (3.8)
	III	**21** (16.0)
	IV	**100** (76.3)
	Unknown	**3** (2.3)

**Table 2 ijms-23-10556-t002:** Sensitivity and specificity of ICC and FISH compared with NGS. FN, false-negative; FP, false-positive; TN, true-negative; TP, true-positive; Se: sensitivity; Spe: specificity.

Technique	ALK						ROS1					
	TP	FP	FN	TN	**Se**	**Spe**	TP	FP	FN	TN	**Se**	**Spe**
ICC	11	7	3	69	**0.79**	**0.91**	3	12	0	78	**1**	**0.87**
FISH	13	0	0	14	**1**	**1**	2	0	0	13	**1**	**1**

## Data Availability

Not applicable.

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
