# Peer review of "Comparison between Immunocytochemistry, FISH and NGS for ALK and ROS1 Rearrangement Detection in Cytological Samples"

_ijms, 2022, doi:10.3390/ijms231810556_

Round 1

Reviewer 1 Report

The manuscript by Frankel et al. evaluates different detection methods for genetic rearrangements in non-small cell lung cancer (NSCLC) patients. The authors analysed 131 samples of patients diagnosed with NSCLC and compared the results for ALK and ROS1 rearrangements on cytological specimens by immunocytochemistry (ICC) and FISH to Next-Generation Sequencing (NGS).

All of the samples were analyzed for ALK and ROS1 rearrangements using NGS, indicating that this is already the method of choice at the authors´ institute, and NGS is also recommended by recent guidelines. Nevertheless, authors provide evidence that use of ICC and, subsequently, FISH is sensitive and specific to detect the gene rearrangements in question. In line with others that were investigating sensitivity and specificity of ICC and FISH to detect ALK and ROS genetic rearrangements they confirm that this combination is sufficient to diagnose these targetable molecular alterations in NSCLC.

Major comments:

 -As the authors´ institution has already committed to using NGS for gene fusion detection in lung cancer, the authors are encouraged to re-focus their main message, i.e. that they evaluate a cost-effective strategy (sequential use of ICC and FISH) for finding therapeutically relevant ALK and ROS aberrations in NSCLC. Then, the take-home is that NGS is the gold standard but combining ICC and FISH is equally sensitive and specific.

-Reports on “additive results” (2.4.), e.g. gene mutations detected by panel sequencing are somehow anecdotal without putting them into context. Instead, the authors could check for co-occurrence of TP53 mutations and ALK rearrangements that have been described by others (e.g. https://pubmed.ncbi.nlm.nih.gov/30165392/) and include results in a matrix describing all mutational events and rearrangements.

-It is not clear what to learn from table 2. Maybe a pie chart would be more intuitive to understand the distribution of samples. Moreover, the numbers do not add to 131 for ICC and detection of ROS1.

Minor comments:

Authors should ensure that they confer the entire information needed to understand what was actually performed, e.g. line 72

ALK and ROS1 rearrangements were both performed on all samples” “hyperploid” (line 106, should probably read “hyperdiploid”) and to reach out to native language editing to avoid ambiguous wording as in “a very few amount of malignant cells can be sufficient” (line 200), “When the amount of material was enough…” (line 257), “avalaible”, instead of “available”, line 80 etc.

The authors might also want to cite a recent report, which comes to similar conclusions regarding ALK rearrangement testing (https://jcp.bmj.com/content/75/6/405).

Author Response

We thank you for these comments that help to improve our manuscript.

You will find below your comments shown in italics our corrections in blue.

Reviewer 2 Report

Dear authors,

Congratulations on the performed hard work to prepare the manuscript.

Your findings show that cytological materials are suitable for molecular analysis. For the detection of ALK and ROS1 rearrangements, results from the ICC and FISH techniques are extremely comparable to those from NGS.

I have only a minor suggestion to improve the quality of your work: maybe some graphics with the data from tables would be useful to visualize the differences between the results obtained by each testing proceduce.

Nevertheless, the results are clearly presented, the materials and statistical methods easy to follow and the discussion well exposed.

Kind regards,

Author Response

(The authors gave the same response as above.)
